# Biological Potential of Polyphenols in the Context of Metabolic Syndrome: An Analysis of Studies on Animal Models

**DOI:** 10.3390/biology11040559

**Published:** 2022-04-07

**Authors:** Joanna Niewiadomska, Aleksandra Gajek-Marecka, Jacek Gajek, Agnieszka Noszczyk-Nowak

**Affiliations:** 1Doctoral School of Wroclaw, University of Environmental and Life Sciences, 50-375 Wroclaw, Poland; 2Department of Cardiology, Kłodzko County Hospital, 57-300 Kłodzko, Poland; olagajek@interia.eu; 3Department of Emergency Medical Service, Wroclaw Medical University, 50-556 Wroclaw, Poland; jacek.gajek@umed.wroc.pl; 4Department of Internal and Diseases with Clinic for Horses, Dogs, and Cats, Faculty of Veterinary Medicine, Wroclaw University of Environmental and Life Sciences, 50-375 Wroclaw, Poland; agnieszka.noszczyk-nowak@upwr.edu.pl

**Keywords:** polyphenols, laboratory animal models, metabolic syndrome

## Abstract

**Simple Summary:**

Polyphenols are a family of widespread organic compounds occurring in heterogeneous plants. The antioxidative properties of polyphenols are widely known. However, they also possess unquestionably anti-inflammatory, antiatherosclerotic, anti-microbial, antiallergic, immunomodulatory, anti-carcinogenic, and antimutagenic activity. As evidenced by recent studies, diets abundant in polyphenol-rich plant materials induce positive effects in preventing and treating non-communicable chronic diseases, such as metabolic syndrome, which is currently recognized as a crucial public health problem. This study aims to review the literature on the role of polyphenols in the aspect of metabolic syndrome in studies on animal models and highlight the most promising micronutrients. Positive results in this field can lead to the development of innovative functional food products, which can contribute to the health improvement of the population.

**Abstract:**

Metabolic syndrome (MetS) is a disease that has a complex etiology. It is defined as the co-occurrence of several pathophysiological disorders, including obesity, hyperglycemia, hypertension, and dyslipidemia. MetS is currently a severe problem in the public health care system. As its prevalence increases every year, it is now considered a global problem among adults and young populations. The treatment of choice comprises lifestyle changes based mainly on diet and physical activity. Therefore, researchers have been attempting to discover new substances that could help reduce or even reverse the symptoms when added to food. These attempts have resulted in numerous studies. Many of them have investigated the bioactive potential of polyphenols as a “possible remedy”, stemming from their antioxidative and anti-inflammatory effects and properties normalizing carbohydrate and lipid metabolism. Polyphenols may be supportive in preventing or delaying the onset of MetS or its complications. Additionally, the consumption of food rich in polyphenols should be considered as a supplement for antidiabetic drugs. To ensure the relevance of the studies on polyphenols’ properties, mechanisms of action, and potential human health benefits, researchers have used laboratory animals displaying pathophysiological changes specific to MetS. Polyphenols or their plant extracts were chosen according to the most advantageous mitigation of pathological changes in animal models best reflecting the components of MetS. The present paper comprises an overview of animal models of MetS, and promising polyphenolic compounds whose bioactive potential, effect on metabolic pathways, and supplementation-related benefits were analyzed based on in vivo animal models.

## 1. Introduction

Metabolic syndrome (MetS), a risk factor for cardiovascular diseases (CVD) and type 2 diabetes, affects a significant part of the population worldwide, with a prevalence of 10–30%. It is a clustering of interrelated metabolic disorders, which include insulin [1,2,3,4,5,6] resistance, central obesity, hypertriglyceridemia, lowered HDL cholesterol concentration, and hypertension [1,2]. MetS has had different criteria over the years, mainly associated with distinguishable definitions of abdominal obesity, with the World Health Organization (WHO), the National Cholesterol Education Program Adult Treatment Panel (NCEP/ATP111), the American Association of Clinical Endocrinologists (AACE), and the European Group for Study of Insulin Resistance (EGSIR) all proposing their own diagnostic criteria [3,4]. Finally, in 2005, the International Diabetes Federation (IDF) provided a standardized consensus. The proposed definition includes waist circumference as a precondition for the identification of MetS and embraces the standard features of the previous definition, such as the assessment of triglyceride (TG) level, high-density lipoprotein cholesterol (HDL), blood pressure, and fasting glucose [5,6].

Metabolic pathways comprising the pathomechanism of MetS have not yet been clearly characterized. However, this is a tedious process due to the wide range of different pathophysiological mechanisms needing to be considered. Evidence suggests that various factors may predispose one to the development of MetS, such as genetics, diet, lifestyle, and gut microbiome [7,8]. A syndrome, which is more of a clinical term than a disease entity, suggests an association with other disorders. The current research shows that MetS predisposes one to cardiovascular dysfunctions via, e.g., atherosclerotic changes [9,10,11] and type 2 diabetes [12]. The correlation with other disorders is based on oxidative stress’s role in the pathomechanism. Studies have indicated that there may be an association between MetS and Parkinson’s disease [13], obstructive sleep apnea [14], and the progression and development of different cancers, such as colon cancer or gastric cancer [15,16,17]. Treatment is mainly based on lifestyle changes involving increased physical activity and a balanced diet. Researchers are currently looking for new substances that could significantly mitigate the severity and progression of MetS symptoms by affecting the metabolic pathways involved in the MetS pathophysiology. Numerous studies based on animal models have demonstrated the existence of a relationship between the intake of polyphenol-rich products and the mitigation of individual components of MetS. A beneficial effect was obtained via a reduction in body weight, blood pressure, blood glucose levels, and improved lipid metabolism.

The aim of this study is to review the contemporary literature on the role of polyphenols in the aspect of metabolic syndrome in studies on animal models, which provide information that cannot be obtained in humans.

### 1.1. Pathophysiology of the Metabolic Syndrome

The main molecular changes in MetS result from a complex interaction between genetic and environmental factors. Visceral adipose tissue endocrine mediation, insulin resistance, and hypertension have been included as pathophysiological elements. The literature also highlights the contribution of endothelial dysfunction, systemic inflammation, and oxidative stress to MetS pathogenesis. However, it is difficult to identify individual pathophysiological mechanisms due to overlapping changes, where one pathology generates the next one, which determines yet another, and so forth.

#### 1.1.1. Obesity

One of the main change-inducing factors is visceral obesity. An excess of adipose tissue contributes to the disruption of the body’s homeostasis and the initiation of adaptive changes [18]. The state of positive energy balance and low-grade chronic inflammation leads to increased plasma FFA levels, which result in ectopic lipid storage and lipotoxicity. It is believed that the accumulation of visceral adipose tissue precedes the development of insulin resistance, and its role in MetS is associated with the secretion of numerous inflammatory mediators. Inflammation, in turn, is inextricably linked to the pathogenesis of atherosclerosis, forming a link between obesity and increased risk of CVD [18]. Adipose tissue as a whole is an endocrine organ. Adipocytes secrete numerous bioactive substances called adipocytokines, which maintain systemic homeostasis. An altered profile of adipocytokines may stimulate the development of MetS and play a crucial role in cellular dysfunction. The cytokines that predominantly contribute to abnormalities are resistin, leptin, adiponectin, TNF-α, and IL-6. A unique role is played by resistin, whose increased secretion by adipocytes in obese individuals correlates with increased cellular insulin resistance [19].

#### 1.1.2. Insulin Resistance

The core element of MetS is insulin resistance (IR). Insulin resistance decreases the ability of various organs, for example, the liver, skeletal muscle, or adipose tissue—to respond to insulin [20]. Insulin regulates a wide range of biological processes by the activation of two crucial post-receptor transduction signaling cascades, PI3K (phosphatidylinositide 3 kinase) and RAS-MAPK (mitogen-activated protein kinase) [21]. PI3K cascade activation is responsible for the insulin effect on the metabolism by adjusting the activity of transcription factors responsible for cell proliferation and apoptosis. This pathway in vascular endothelium enhances nitric oxide production, inducing vasodilatation [22,23]. The RAS-MAPK signaling cascade plays a role in cell growth and proliferation and results in vasoconstriction [24,25]. The molecular alterations caused by insulin resistance are based on the downregulation of the PI3K pathway and the upregulation of RAS-MAPK [26,27]. In MetS, the insulin resistance that seems to emerge from positive energy balance is mainly caused by the oxidative stress to which cells are exposed. An excessive intake of energy-providing substances leads to an increased synthesis of nicotinamide adenine dinucleotide phosphate (NADP), which promotes the biosynthesis of reactive oxygen species (ROS). As a means of defense, cells block the entry of energy-providing compounds, including, for example, glucose. Therefore, it may be concluded that insulin resistance in MetS serves as an adaptive mechanism protecting cells from potential damage related to the excessive generation of free radicals, thus exacerbating pathological changes in carbohydrate metabolism [2,19,28].

#### 1.1.3. Free Radicals

Free radicals are small, diffusible, and highly reactive molecules marked by cytotoxic and genotoxic effects. The production of reactive oxygen species (ROS) is many associated with dysfunctional homeostasis, though some of these, called bioradicals, originate from the physiological process [17,29]. The excess accumulation of free radicals leads to chronic inflammation and an imbalance in cellular apoptosis and proliferation via the altered hyper- or hypo-activation of some cellular signaling pathways [17]. In MetS, visceral obesity contributes to the overproduction of adipokines [30,31]. Abnormal adipokine levels yield a persistent increase in systemic inflammation, and the infiltration of macrophages in visceral adipose tissue has collectively been indicated as a possible factor enhancing reactive oxygen species production. ROS contribute directly to autonomic balance dysregulation and, in turn, to inadequate blood pressure control [32,33].

#### 1.1.4. Renin–Angiotensin–Aldosterone System (RAA)

The cardiovascular system is also subject to alterations related to MetS. Essential roles in hemodynamic pathophysiology are played by the activation of the renin–angiotensin–aldosterone system [34]; differing levels of adipocytokine secretion—i.e., leptin, tumor necrosis factor (TNF-α), and interleukin 6 (IL-6) [35,36]; and the hyperactivity of the sympathetic nervous system [37,38]. The hyperactivity of the sympathetic nervous system alone contributes to an increase in heart rate, circulating blood volume, ventricular end-diastolic volume, and cardiac output, which can directly—or indirectly via a feedback loop with the RAA system—lead to the development of hypertension [10]. The activation of the RAA system in the insulin resistance state is closely related to sodium retention, which leads to increased intravascular volume. However, an increase in the aldosterone serum level may also occur due to the upregulation of the angiotensinogen gene in adipose tissue. Studies suggest that increased proinflammatory adipokine secretion contributes to RAA activation by stimulating angiotensinogen production in adipocytes. The local stimulation of RAA in visceral adipose tissue may be critical in the pathogenesis of hypertension in obesity and metabolic syndrome [34,39].

### 1.2. Cardiovascular Consequences

The changes that occur in the cardiovascular system are extensive and lead to cardiomyopathy, microcirculation damage, and endothelial function impairment [9,10]. Each component of MetS is an independent risk factor for cardiovascular diseases. Studies indicate an association between MetS and the elevated risk of atherosclerosis, myocardial infarction, and heart failure. Obesity-associated cardiomyopathy is characterized by concentric left ventricular hypertrophy and systolic or diastolic dysfunction. In addition, myocardial contractility, systolic velocity, and left ventricular shortening have been proven to be impaired [40,41]. Changes in microvascular tone and density are attributable to the non-equilibrium between oxygen delivery and tissue metabolism in miscellaneous vascular beds. These alternations in MetS are mainly caused by the significant variations existing in the control of arteriolar resistance [42]. Endothelial dysfunction is associated with decreased nitric oxide bioavailability in the setting of MetS. The underlying mechanism contributing to endothelial pathology is the augmented production of vasoconstrictors, including endothelin-1 (ET-1), thromboxane A_2_ (TXA_2_), and prostaglandin H_2_ (PGH_2_) [42]. The progression of atherosclerosis occurs over the years and is strongly correlated with age. The relation between the metabolic abnormalities occurring in MetS and atherosclerotic disease is undoubted. MetS leads to accelerated and more advanced atherosclerotic disease, which is correlated with a greater incidence of myocardial infractions. The adipose-derived hormones and adipokines released from fat depots, including perivascular adipose tissue, are considered to be the core of the pathological process and thought to mediate vascular calcification [43,44].

### 1.3. Polyphenols

Polyphenols are the most widespread bioactive compounds derived from plants. The basic monomer forming these secondary metabolites is a phenolic ring. According to their diverse chemical structures, polyphenols are classified into two major groups, flavonoids and nonflavonoids, as well as many subsequent groups [45,46]. The most widely known polyphenol substances include phenolic acids, flavonoids, stilbenes, lignans, and phenolic alcohols. Fruits and beverages constitute the core sources of polyphenolic compounds. Plants contain mixtures of polyphenols. They are considered to play a crucial role in adapting plants to their environment. In addition, they represent a significant source of bioactive pharmaceuticals [47]. Their health-promoting properties are mainly attributed to their antioxidant activity. However, polyphenols also possess pronounced anti-inflammatory, antiatherosclerotic, antiallergic, anti-microbial, anti-carcinogenic, and antimutagenic activities [48]. Considering that chronic progressive inflammation is a feature of MetS, polyphenols appear to be promising dietary supplements for preventing the progression of the disease and minimizing the effects of MetS.

### 1.4. Usability of Animal Models in MetS Research

As the MetS pathophysiology is involved plenty of genetic variations and environmental factors, it is troublesome to discover one by one particular underlying components in such a complex system as a human being. Scientists have established many animal models that mimic the metabolic complications in humans to look for a solution. They are fundamental in discovering the genetic fundamentals of diverse parameters, characteristic of MetS, and their dependence on each other. Given the same environment, we can explore how certain conditions impact metabolic changes and analyze the effect of environment on phenotype. In vivo models render genetic and environmental factors controlled, adjusted, and monitored in organisms of known origin. Understanding pathophysiology in animal models is crucial for implementing appropriate therapies and prevention strategies. Animal subjects, providing stable conditions, are thought to be small-scale follow-up research embracing relevant human studies [49,50]. 

## 2. Methods

Numerous studies on animal models have been conducted to identify the mechanisms of polyphenols’ impact on biochemical pathways in MetS. Multiple compounds have been investigated, including quercetin, epigallocatechin gallate, naringenin, resveratrol, and extracts obtained from polyphenol-rich foods. The study aims to point out the natural polyphenols or polyphenols themselves that are of the utmost importance on the course of the disease. Studies on selected animal models mirroring the components of MetS were taken into consideration.

### 2.1. Search Strategy

This research was carried out in the PubMed and ScienceDirect search libraries. The search keywords used in PubMed were: “selected animal model” and “metabolic syndrome” and (“polyphenols” or “phenols” or “flavonoids” or “flavonols”). The search terms used in ScienceDirect were as follows: “selected animal model” and “polyphenols” and “metabolic syndrome”; filtered: years: 2000–2021; article type: research articles. For each animal model, the search in both databases was conducted separately. Only papers in English were analyzed. The time interval considered comprised the years 2000 to 2021.

### 2.2. Inclusion Criteria

The research performed on preferable animal models in the context of metabolic syndrome (Zucker Fatty Rat (fa/fa), Zucker Diabetic Fatty Rat, Spontaneously Hypertensive Rat (SHR), animal models with induced diabetes/obesity/hypertension (pathophysiological changes playing an essential role in metabolic syndrome)).The presence of a control group and the comparison between the polyphenol’s intervention group and the placebo group were incorposed. The administration route and the dosage of polyphenols were not restricted.

### 2.3. Exclusion Criteria

Studies without indicating a statistically significant difference between the control and experimental group.Papers with incomplete data and duplicated publications.Studies on animal models differed from those mentioned in the inclusion criteria.

The studies were selected from 695 publications found. Two independent researchers reviewed all eligible papers (JN and AGM). Based on the results obtained for each animal model, the most promising substances were chosen. 

## 3. Animal Model of Obesity Zucker Fatty Rats (fa/fa)

Laboratory ZF rats are used in human disease studies as a model of obesity with accompanying hyperlipidemia and hypertension. While this model is most widely used in studies on genetic obesity, ZF rats are also used in studies on MetS and non-insulin-dependent obesity-related diabetes. ZF rats are characterized by a recessive mutation in the leptin receptor gene (called “fa”), which leads to polyphagia, with the consequent development of obesity at around four weeks of age. The causes of obesity in ZF rats also include hypertrophy and adipocyte hyperplasia, which are linked to their genetic predisposition. Other conditions observed in ZF rats include hyperinsulinemia and impaired glucose tolerance, which do not lead to overt diabetes [51,52].

Studies conducted using this animal model have shown that many polyphenolic substances have potentially beneficial metabolic effects in extracts or individual compounds. 

### 3.1. Red Wine

Grapes and red wine are rich sources of phenolic acids, flavonols, quercetin, (+)-catechin, dihydroflavonols, anthocyanins, catechins, and stilbenes [53,54]. Red wine polyphenols were first noticed as very useful with the identification of the theory called the “French Paradox”. This theory pointed out that the high amount of red wine polyphenols consumed by the French every year is responsible for the comparatively low level of coronary heart disease (CHD) among the French population [55,56]. The French concept later became the reason for investigating the role of red wine constituents as cardioprotective factors. Following these findings, scientific studies proved their protective action in the vascular system. Moreover, compounds from red wine protect against cerebrovascular incidents [57]. Additionally, in vitro studies evidenced that the supplementation of red wine polyphenols reduced inflammation and NADPH oxidase activity and increased endothelial nitric oxide production [58,59]. In vivo, animal studies seem to confirm these findings. An animal model study investigated dietary supplementation with red wine polyphenol extract on metabolic, circulatory, and vascular changes. The analysis found that the polyphenol extracts improved glucose metabolism by reducing serum glucose levels and improved lipid profiles by lowering triglyceride and LDL cholesterol levels. In turn, echocardiographic measurements showed an increase in fractional shortening and cardiac output. The analysis also showed an increase in nitric oxide (NO) bioavailability associated with increased endothelial NO-synthase (eNOS) activity and, consequently, a reduction in peripheral arterial resistance. In turn, the decreased expression of NADPH oxidase inhibited the release of superoxide anions [60]. The decline in vascular tone is probably linked to the modulation of the expression of cyclooxygenase (COX) and COX-derived vasoconstrictive agents via a mechanism that involves the NF-κB pathway. The vasoprotective effect of the dietary supplementation of red vine polyphenols is also associated with a reduction in the release of vasoconstrictive factors such as thromboxane-A2 and 8-isporostone [61,62].

### 3.2. Green Tea

Green tea is a rich source of catechins, including epigallocatechin gallate (EGCG), an organic chemical compound belonging to the polyphenol family. Green tea extract, as well as EGCG alone, were analyzed in studies on ZF rats to verify their impact on weight, lipid profile, and glucose metabolism. One study investigated the protective effects and molecular mechanisms of action of green tea polyphenols in non-alcoholic fatty liver disease (NAFLD). In that study, pathological metabolic changes in hepatocytes identical to those seen in humans with NAFLD were induced in ZF rats by a high-fat diet. A decrease in body weight and a statistically significant reduction in visceral fat (31.0%, *p* < 0.01) were observed in rats treated with green tea polyphenols compared with controls. Moreover, significant decreases in fasting insulin, glucose, and lipid levels were observed. The observed reduction in hepatic lipogenesis was linked to the upregulation of the AMPK pathway [63]. In another study, an intraperitoneal injection of green tea catechin extract, mainly containing EGCG, was found to reduce food intake and body weight and lead to a number of changes in the endocrine system, including a reduction in the blood levels of testosterone, estradiol, leptin, insulin, IGF-1, LH, glucose, cholesterol, and triglycerides. This experiment was performed on Sprague Dawley and Zucker Fatty rats. Similar effects were observed in both groups, suggesting that the effect of EGCG on appetite control is independent of leptin. The effective dose of EGCG was approximately 30–50 mg/kg BW. The loss in body weight was reversible. When the administration of EGCG was stopped, the rats regained their weight [64]. The beneficial effect of green tea polyphenols on weight gain attenuation, the reduction in visceral fat accumulation, and the decline in insulin level and fasting serum glucose may be associated with molecular changes in the expression of insulin signaling protein in skeletal muscle. The polyphenols from green tea administered to Zucker Fatty (ZF) rats fed a high-fat diet at a dose of 200 mg/kg of body weight for 8 weeks resulted in lower insulin resistance. Immunoblotting revealed that the expression and translocation of glucose transporter-4 were enhanced in skeletal muscle. The insulin-stimulated glucose uptake by isolated muscle in ZF rats treated with green tea polyphenols increased as well. Moreover, a decrease in the activation of the inhibitory protein kinase isoform, PKC-θ, which is muscle-specific, was also observed. This outcome shows that the effects of polyphenols from green tea may be associated with the impact on skeletal muscle insulin sensitivity [65]. The ingestion of green tea polyphenols can ameliorate the metabolic abnormalities linked with MetS and promote favorable molecular effects.

### 3.3. Quercetin

Quercetin is a bioflavonol found in numerous plant foods, such as beans, red onion, lettuce, broccoli, citrus, tea, wine, and herbs. It is believed that quercetin has significant antioxidant potential and thus shows protective effects concerning osteoporosis, cardiovascular disease, neuropathy, and even some types of cancer, for example, breast cancer, lung cancer, and colon cancer [66,67,68,69,70]. Moreover, it is characterized by anti-inflammatory, antiobesity, antihyperlipidemic, antihypercholesterolemic, neuroprotective, antihypertensive (via vasodilator effects), and antiatherosclerotic properties [67,71,72]. The role of quercetin in organism metabolism is assumed to be mediated via the activation of transcription factors such as PPAR-γ, AMPk, NF-κB, or SIRT1 [69,73]. A daily quercetin dose of 2 mg/kg BW or 10 mg/kg BW administered to obese ZF rats for ten weeks resulted in reduced dyslipidemia, hypertension, and insulin resistance. However, only the use of the higher dose reduced body weight and produced anti-inflammatory effects by lowering TNF-alpha production in visceral adipose tissue. The decline in body weight was associated with an increase in the plasma concentration of adiponectin, reduced levels of which are observed in obesity, type 2 diabetes, and hypertension. The vasoprotective effects of quercetin were shown to be mediated by an increase in eNOS expression [74].

## 4. Animal Model of Diabetes: Zucker Diabetic Fatty (ZDF) Rats

Laboratory ZDF rats are derived from the Zucker Fatty strain. A spontaneous mutation that occurred in ZF rats resulted in a diabetic phenotype. The inbreeding of ZF rats carrying the desired mutation led to the development of a new strain called the Zucker Diabetic Fatty strain. The cause of the genetic defect leading to impaired beta-cell function is not clear. However, a number of changes in the expression of pancreatic islet genes have been described, including a reduced expression of the GLUT2 transporter and increased activity of glucokinase and hexokinase. ZDF rats have higher insulin resistance and are less obese compared with the parental strain [51]. Male ZDF rats develop diabetes at eight weeks of age, which is associated with changes in the morphology of pancreatic islets. Female rats do not develop overt diabetes, despite their significant level of insulin resistance, except when they are fed a high-fat diet [75]. In male ZDF rats, diabetes-related cataract is observed at 15 weeks of age, first as angiogenesis changes at the periphery of the lens, which progress to the development of a mature cataract at 21 weeks of age [76,77]. Laboratory ZDF rats provide, in particular, a suitable animal model of type 1 and type 2 diabetes and its complications, including retinopathy, cardiomyopathy, and diabetic nephropathy. In addition, due to its specificity (hyperglycemia, hyperinsulinemia, impaired glucose metabolism), the strain can be used in studies on MetS [49].

The polyphenolic compounds analyzed in studies conducted using this animal model include pomegranate extracts and cocoa flavonols.

### 4.1. Pomegranate

The pomegranate (Punica granathum L.) is a long-lived plant that belongs to the Lythraceae family. Its various parts are rich sources of a variety of compounds. Pomegranate seed oil contains punicinic acid (a polyunsaturated fatty acid) and phytoestrogens. The juice and peel are rich in numerous polyphenolic compounds, especially tannins and flavonoids. Pomegranate tannins include ellagitannins, such as punicalagin and punicalin, whereas pomegranate flavonoids include, in particular, anthocyanins and flavonols. Moreover, pomegranate juice and peel contain numerous catechins. In turn, pomegranate bark and roots are sources of alkaloids, including pelletierine and isopelletierine, which are used in folk medicine as anthelmintics. Pomegranate polyphenols have antioxidant properties, as they indirectly inhibit inflammatory markers. They also present anti-carcinogenic effects [78,79]. Pomegranate extracts seem to have a beneficial effect on changes characteristic of MetS, as confirmed by the findings from the available studies. ZDF rats treated with pomegranate flower extract (500 mg/kg, p.o. × six weeks) had milder symptoms of diabetes- and obesity-related hepatic steatosis (lower liver weight, lower triglyceride levels, and lower lipid droplet content) compared with controls. This was due, at least in part, to the enhanced expression of genes associated with the oxidation of fatty acids, including peroxisome proliferator-activated receptors (PPAR-α), acyl-CoA oxidase, and carnitine palmitoyltransferase-1 [80]. A 6-week treatment with pomegranate extract (500 mg/kg p.o.) was also found to reduce the collagen deposit area in the left ventricle as well as the perivascular collagen deposit areas. The reduction in cardiac fibrosis was mediated by the modulation of the endothelin-1 (ET-1) and nuclear factor kappa B (NF-κB) pathways. The diminished cardiac fibrosis was accompanied by reduced hyperglycemia and hyperlipidemia [81]. In another study on ZDF rats with insulin resistance and hyperlipidemia, *Punica granathum* extract reduced cardiac triglyceride accumulation and decreased circulating triglyceride and cholesterol levels. In that study, the improvement in cardiac lipid metabolism was mediated by the activation of, e.g., PPAR-alpha [82].

### 4.2. Cocoa

The cocoa tree (*Theobroma cacao* L.) is a tree in the family *Malvaceae* native to the forests of South and Central America. Its seeds, commonly known as cocoa, are used in many food products. The chemical composition of cocoa includes numerous polyphenolic compounds, including proanthocyanins, catechins, flavan-3-ols, and anthocyanins. It is currently being investigated whether it may play a significant role as a dietary intervention to reduce cardiovascular risk in type 2 diabetes, given that it reduces plasma lipid levels and promotes the production of nitric oxide (NO) [83]. Recent studies have shown that cocoa polyphenols also have beneficial effects on carbohydrate metabolism. In one study, male ZDF rats fed a cocoa powder-rich diet (10%) for ten weeks showed improved glucose tolerance and insulin resistance. Moreover, the consumption of a diet rich in cocoa products had protective effects against diabetes-induced structural alterations in the kidneys. The antihyperglycemic effects of cocoa, protecting against diabetic nephropathy, were mediated through the inhibition of the synthesis of gluconeogenic enzymes, i.e., phosphoenolpyruvate-carboxykinase (PEPCK) and glucose-6-phosphatase (G-6-Pase), and glucose transporters (i.e., sodium-glucose-co-transporter-2 (SGLT-2) and glucose-transporter-2 (GLUT-2)) in the renal cortex [84]. In both in vivo (ZDF rats) and in vitro (HepG2 cells) models, cocoa flavonols, especially (−)-epicatechin, improved lipid metabolism by reducing body weight gain and lipid accumulation in liver cells [85]. Cocoa intake also improves the gut microbiota via interactions that may contribute to its antidiabetic effect. Male ZDF rats fed with 10% cocoa presented more acetate-producing bacteria and had a reduced amount of lactate-producing bacteria compared to the lean group. The modified gut microbiota was associated with an improvement in glucose homeostasis and intestinal integrity and with a reduced expression of pro-inflammatory cytokines, such as tumor necrosis factor-a (TNF-a) or interleukin-6 (IL-6), in the colon of rats [86].

## 5. Animal Model of Hypertension: Spontaneously Hypertensive Rat (SHR)

SHR rats were produced by outbreeding Wistar–Kyoto rats, followed by the selective inbreeding of their offspring with the highest blood pressure values. These rats are mainly used in studies on cardiovascular diseases, especially hypertension, metabolic diseases that lead to insulin resistance, hypertriglyceridemia, hyperinsulinemia, hypercholesterolemia, and renal dysfunction. SHR rats develop hypertension at 6–7 weeks of age, reaching a stable level with accompanying insulin resistance and hyperinsulinemia at 12 weeks of age. The development of hypertension in these rats is probably mediated by the renin–angiotensin axis and the activity of the sympathetic nervous system [49,51]. SHR rats do not develop hypercholesterolemia and hyperlipidemia, except where a specific dietary regimen induces these changes. For many years, these rats have been used in studies on heart failure due to the changes that occur in their cardiac muscle, including the progressive hypertrophy of the left ventricle, which is evident over the first 9 months of life and progresses to systolic dysfunction by 12 months of age. A variation of SHR rats is SHR stroke-prone (SHRSP) rats. SHRSP rats develop malignant hypertension and die from stroke within a few weeks [51].

As this animal model is well suited for assessing the impact of a given substance on the cardiovascular system, one of the most exciting polyphenols studied using SHR rats is resveratrol, a compound with cardioprotective properties. The characterization of metabolic changes in genetic animal models of metabolic syndrome is presented in Table 1.

### Resveratrol

Resveratrol is an organic chemical compound that is a member of a polyphenol family called viniferins. It is present in numerous plants, such as eucalyptus, mulberry, peanut, bilberry, strawberry, grape, rhubarb, and cranberry [87]. Grapes are considered one of the most abundant sources of resveratrol, as they have the highest levels of this compound, with their concentration in peel and seeds reaching 50–100 µg/g [88]. This stilbene derivative exhibits significant biological potential. In natural conditions, resveratrol is produced by some plants in response to injury. The available studies on this topic have shown that it has estrogenic, anti-inflammatory, cerebroprotective, cardioprotective, anti-angiogenic, antioxidant, and anti-cancer properties [88,89]. The effects of resveratrol are concentration-dependent, as it acts as a chemoprotective agent at a specific dose and promotes apoptotic cell death when used at a higher concentration [90]. In one study, the administration of resveratrol in drinking water to SHR rats for 10 weeks reduced the development of hypertension, as indicated by the lower blood pressure of the treated rats compared with the controls. Resveratrol-treated rats showed reduced hydrogen peroxide (H_2_O_2_) content and increased superoxide dismutase activity, thus reducing their oxidative stress. Moreover, the rats displayed the normalization of endothelium-dependent vasorelaxation [91]. Resveratrol may also prevent hypertension-induced cardiac dysfunction. In one experiment, a ten-week treatment with resveratrol significantly reduced concentric hypertrophy and systolic dysfunction in SHR rats. The cardioprotective effects of resveratrol are probably mediated by a lowering of oxidative stress levels in the cardiac muscle tissue [92]. Resveratrol may also play a role in electrophysiological alternations via its influence on chromaffin cells and Ca^2+^ signaling, exerting antihypertensive effects through these mechanisms. Systolic blood pressure decreased in a group of SHR rats exposed to trans-resveratrol treatment in drinking water (50 mg/L/v.o.) for 28 days. However, no reversal of cardiac hypertrophy was observed. The study exhibited an increase in outward voltage-dependent potassium currents (I_K_), a reduction in inward voltage-dependent sodium (I_Na_), calcium (I_Ca_), and nicotinic (I_Ach_) currents, and an attenuation of cytosolic Ca^2+^ concentration overload in chromaffin cells from SHR rats. Based on these findings, the modulation of the sympathoadrenal axis functionality may be a new target that could account for resveratrol’s antihypertensive effect [93].

## 6. Animal Models with Induced Diabetes/Obesity/Hypertension

Animal models with a relevant genetic setup are not the only way to analyze the biological effects of polyphenolic compounds. Pathophysiological changes typical of MetS may also be induced by dietary manipulation or the administration of drugs. The nutritional approaches studied involved administering a single type of diet or a combination of diets to modify metabolic pathways, especially those related to carbohydrate and lipid metabolism, to induce changes that best reflect those observed in people with MetS. To cause hypertension, obesity, hyperglycemia, or dyslipidemia in laboratory animals, they can be fed a diet including large doses of carbohydrates, including fructose and sucrose, or a high-fat diet. The percentage content of carbohydrates or fat in a diet necessary to induce relevant effects varies. For example, in one study, the dose of fructose used to cause hypertension, insulin resistance, and glucose intolerance exceeded 60% of caloric intake [94], whereas the administration of 30% sucrose solution to male Wistar rats was sufficient to induce hypertension as well as an increase in body weight, insulin content, and total lipids [95]. The fat content used in the experiments ranged from 20% to 60% of the total energy demand [96,97]. The most commonly used strains in diet-induced models of MetS are Sprague Dawley rats, Wistar rats, C57BL/6 J mice, and Syrian hamsters [49]. MetS can also be induced in laboratory animals using drugs such as glucocorticoids. In medicine, glucocorticoids comprise the primary treatment for various conditions, including autoimmune disorders, dermatological conditions, and cancer. However, they have certain side effects, which determine their usefulness in triggering a cascade of changes leading to the development of MetS in laboratory animals. Glucocorticoids act on different tissues and organs by, e.g., stimulating the differentiation of preadipocytes into mature adipocytes; increasing lipolysis, glucose intolerance, and body weight gain; and disturbing calcium metabolism. Experiments involving animal models use the effects of both exogenous and endogenous glucocorticoids [98,99].

Studies investigating the properties of various polyphenols have used animal models with induced changes characteristic of MetS. These polyphenols include cinnamon compounds and curcumin. Table 2 summarizes the beneficial effects on metabolic changes by propitious polyphenolic compounds gathered in this paper. 

### 6.1. Cinnamon

Cinnamon is primarily known as a spice obtained from the bark of the Cinnamomum tree. It has numerous medicinal properties. Different parts of the plant are enriched in different chemicals, including eugenol, cinnamaldehyde, camphor, and numerous polyphenols [100]. Cinnamon has been found to present anti-inflammatory, anti-microbial, antiviral, antifungal, antioxidant, cardioprotective, hepato-protective, analgesic, wound-healing, and epithelialization-promoting effects, as well as many other properties [100,101]. Cinnamon compounds also have an important impact on carbohydrate metabolism. The supplementation of a diet with cinnamon reduces insulin resistance due to the tannin content by increasing the expression of PPAR-α and PPAR-γ and stimulating the β-subunits of the insulin receptors of adipocytes [102,103]. In vivo studies on animals seem to confirm that cinnamon compounds also have beneficial effects on MetS. A study on Wistar rats fed a high-fat/high-fructose diet for 12 weeks found that a diet containing 20 g of cinnamon improved insulin sensitivity and reduced peritoneal fat accumulation without achieving a statistically significant reduction in body weight compared with controls. The improved insulin sensitivity is probably mediated mainly by the trimeric and tetrameric type A polyphenols present in cinnamon [104]. The wide range of biological activities of cinnamon was confirmed in a study on male Sprague Dawley rats, in which obesity was induced by a high-fat diet, whereas diabetes was induced by the subcutaneous injection of alloxan. A reduction in body weight and fat mass and a decrease in serum leptin levels were observed in rats whose diet included cinnamon extract. Moreover, the administration of cinnamon extract resulted in normalized levels of liver enzymes and reduced blood glucose levels. Furthermore, it produced a dose-dependent antioxidant effect [105]. The findings of many in vivo studies on animals allow the hypothesis that supplementation with cinnamon also has a beneficial impact on MetS [106].

### 6.2. Curcumin

Curcumin is a polyphenolic compound that is naturally present in turmeric rhizomes. It has anti-inflammatory, anti-carcinogenic, and antioxidant properties. Moreover, it shows antibacterial, antiviral, and antifungal effects. Its mechanism of action inhibits the expression of the NF-kB transcription factor, which in turn regulates the expression of numerous proteins involved in the initiation and maintenance of inflammation, which underlies multiple conditions [107]. In a study on male albino Wistar rats, in which a high-fat diet induced diabetes with a dose of streptozotocin, the administration of curcumin for 8 weeks (80 mg/kg BW/day) lowered glucose levels and reduced insulin resistance, dyslipidemia, and lipid peroxidation. Moreover, the administration of curcumin significantly increased the expression of the GLUT-4 gene, which regulates insulin-dependent glucose transport in muscles and adipose tissue, compared with the control group. The regulation of this transporter is altered under pathological conditions, including, among others, type 2 diabetes. The administration of curcumin was found to stimulate the expression of GLUT4, thus normalizing glucose metabolism in the treated group [108]. Curcumin can also be used as a dietary intervention against lipid accumulation and liver fibrosis. It acts via the stimulation of lipogenic gene expression and, in this way, induces lipolysis and inhibits lipogenesis. In groups of Wistar rats administered curcumin at a dose of 100 mg/kg for 4 weeks, lipid imbalance was induced by bile duct ligation. A reduction in hepatic fat accumulation via AMPK upregulation was observed. AMPK is a serine/threonine-protein kinase that is responsible for lipid metabolism. Its dysregulation may lead to the development of hepatic injury. Curcumin seems to improve the expression of AMPK and hepatic redox potential and attenuate lipid peroxidation. A curcumin-treated group also showed protective effects against hepatic fibrosis. The hepatic protection was also associated with a reduction in the lipid level in serum by curcumin [109].

**Table 2 biology-11-00559-t002:** The effects of selected polyphenolic compounds with promising bioactive potential on MetS.

Substance	Animal Model	Dose (Time)	Metabolic Effect	Mechanism	References
Red wine	Zucker Fatty rats	20 mg/kg BW (8 weeks)	↑ FS ↑ CO ↓ serum glucose ↓ LDL cholesterollevel ↓ TG level ↓ peripheral arterial resistance ↓ superoxide anions ↓ thromboxane A2 ↓ 8-isoprostane	↑ NO bioavailability ↑ eNOS activity ↓ NADPH oxidase expression	[60]
Green tea	Zucker Fatty rats	200 mg/kg BW (8 weeks)	↓ body weight ↓ visceral fat ↓ hepatic lipogenesis ↓ insulin level ↓ glucose level ↓ lipids level	↑ expression AMPK-Thr172 ↑ expression phosphorylated acetyl-CoA carboxylase (ACC) ↑ sterol regulatory element-binding protein 1c (SREBP1c)	[63]
	Zucker Fatty rats, Sprague Dawley rats	15 mg, 20 mg, 40 mg (7 days, 4 days)	↓ food intake ↓ testosterone level ↓ estradiol level ↓ LH level ↓ leptin level ↓ insulin level ↓ glucose level ↓ IGF-1 level ↓ cholesterol level ↓ TG level	↓ food intake (hypothalamic neuropeptide gene expression alternation?, changes in bilirubin, alkaline phosphatase activity?)	[64]
	Zucker Fatty rats	200 mg/kg BW (8 weeks)	↓ body weight ↓ visceral fat ↓ insulin level ↓ glucose level ↓ insulin resistance	modulation of insulin signaling protein in skeletal muscle ↑ expression and translocation of GLUT-4 in skeletal muscle ↓ activation of the inhibitory protein kinase isoform- PKC-θ	[65]
Quercetin	Zucker Fatty rats	2 mg/kg BW 10 mg/kg BW (10 weeks)	↓ dyslipidemia ↓ hypertension ↓ insulin resistance ↓ weight (only dose 10mg/kg BW) + anti-inflammatory effect	↑ eNOS expression ↑ adiponectin level in plasma ↓ TNF-alpha production in visceral tissue	[74]
Pomegranate	Zucker Diabetic Fatty rats	500 mg/kg BW (6 weeks)	↓ TG level ↓ lipid droplet content in liver	↑ expression PPAR-α ↑ expression acyl-CoA oxidase ↑ expression CPT1	[80]
	Zucker Diabetic Fatty	500 mg/kg BW (6 weeks)	↓ hyperglycemia ↓ hyperlipidemia ↓ cardiac fibrosis	↓ NF- κB activation in macrophages ↓ expression ET-1	[81]
	Zucker Diabetic Fatty	500 mg/kg BW (6 weeks)	↓ cardiac TG accumulation ↓ TG level ↓ cholesterol level	↑ cardiac expression PPAR-α ↑ cardiac expression CPT-1 ↑ cardiac expression ACO ↑ cardiac expression AMPKαK ↓ cardiac expression acetyl-CoA carboxylase (ACC)	[82]
Cocoa	Zucker Diabetic Fatty	10% cocoa-rich diet (10 weeks)	↑glucose tolerance ↓ body weight ↓ insulin resistance ↓ glucose level ↓ insulin level + nephroprotective effect	↓ renal synthesis PEPCK ↓ renal synthesis G-6-P ↓ expression of glucose transporters (SGLT-2, GLUT-2) in the renal cortex	[84]
	Zucker diabetic Fatty	10% cocoa-rich diet (9 weeks)	↓ body weight ↓ lipid accumulation in liver cells	↑ phosphorylated AMPK level in liver ↑ phosphorylated protein kinase B (AKT) level in liver ↓ phosphorylated protein kinase C (PKCζ) level in liver	[85]
	Zucker diabetic Fatty	10% cocoa-rich diet (10 weeks)	↑ glucose homeostasis ↑ intestinal integrity + modification of gut microbiota	↓ amount of lactate- producing bacteria ↓ expression TNF-α ↓ expression IL-6	[86]
Resveratrol	Spontaneously Hypertensive rats	dissolved in drinking water (concentration 50 mg/L), ad libitum (10 weeks)	↓ hypertension ↓ oxidative stress	↓ H_2_O_2_ content ↓ SOD activity ↓ eNOS uncoupling ↓ NO scavenging	[91]
	Spontaneously Hypertensive rats	2.5 mg/kg BW (10 weeks)	↓ concentric heart hypertrophy ↓ systolic heart dysfunction	↓ oxidative stress in cardiac muscle tissue	[92]
	Spontaneously Hypertensive rats	50 mg/kg BW (28 days)	↓ SBP	↑ outward voltage-dependent potassium currents (I_K_) ↓ inward voltage-dependent sodium currents (I_Na_), ↓ inward voltage-dependent calcium currents (I_Ca_) ↓ inward voltage-dependent nicotinic currents (I_Ach_)	[93]
Cinnamon	Wistar rats (high-fat/high-fructose diet)	20 g cinnamon-rich/kg of diet (12 weeks)	↓ insulin resistance ↓ peritoneal fat accumulation	↑ peroxisome proliferators-activated receptors activity?	[104]
	Sprague Dawley rats (high-fat diet + subcutaneous injection of alloxan)	200 mg/kg BW 400 mg/kg BW (6 weeks)	↑ HDL cholesterol level ↓ body weight ↓ LDL cholesterol level ↓ leptin level ↓ glucose level ↓ liver enzymes levels + antioxidant effect	↓ the intestinal absorption of cholesterol? ↓ appetite? ↓ oxidative stress?	[105]
Curcumin	Wistar rats (high-fat diet + streptozotocin)	80 mg/kg BW (8 weeks)	↓ glucose level ↓ insulin resistance ↓ lipid level ↓ lipid peroxidation	↑ expression GLUT-4	[108]
	Wistar rats (bile duct ligation)	100 mg/kg BW (4 weeks)	↓ hepatic fat accumulation ↓ lipid peroxidation ↓ hepatic fibrosis	↑expression AMPK ↑expression CPT-1a	[109]

## 7. Conclusions

Currently, over one-third of the world’s population is obese. The prevalence rates of hypertension, dyslipidemia, and insulin resistance are similarly high. With the concurrence of these conditions becoming increasingly common, MetS affects a growing number of people. Thus, it is necessary to find new compounds that may mitigate its symptoms and prevent the progression of this disease, as well as develop animal models that closely mirror all the changes characteristic of MetS.

The establishment of animal models with the desired metabolic changes is crucial in order to understand the molecular mechanisms of MetS. However, the advantage of MetS animal models is that they allow the impact of new compounds on morphological, histological, biochemical, and functional changes to be investigated. In contrast, such a broad scope of monitoring is practically impossible in human studies. The clarification of the underlying mechanisms of MetS may result in the development of more effective therapies or dietary interventions, possibly based on natural compounds, in the future.

Researchers currently place their hopes in polyphenolic compounds naturally present in various plants. Numerous substances contained in tea, berries, grapes, blueberries, and many other foods have potential ameliorating effects on particular components of MetS. In vivo studies on animals confirm that the dietary intake of specific polyphenols or their complex mixtures reduces MetS symptoms. However, it is still necessary to carry out relevant analyses on humans. Despite the effects observed in in vivo human studies, dietary interventions have shown that some compounds do not have good bioavailability, leaving their usefulness in question. The development of appropriate preparations could make it possible to overcome this obstacle. Further studies may provide a better understanding of the impact of plant polyphenols on changes characteristic of MetS, and animal models, which undoubtedly most accurately reflect these changes, are necessary to enhance our knowledge of these compounds.

In conclusion, the findings presented in this paper show that there is no universal compound that can improve all pathological effects. Most of them ameliorate effects on the lipid and glucose profiles (red wine, green tea, quercetin, pomegranate, cocoa, cinnamon, curcumin). The consumption of resveratrol and red wine can decrease hypertension. Moreover, additional effects are associated with hepatoprotection (pomegranate, green tea, cocoa, cinnamon, curcumin), nephroprotection (cocoa), cardioprotection (red wine, pomegranate, resveratrol), the modulation of the gut microbiota (cocoa), anti-inflammatory effects (quercetin), and antioxidant effects (resveratrol, cinnamon). The administration of individual polyphenols leads to particular changes, while the use of a combination of related polyphenols may provide synergistic effects and lead to more significant benefits. Further studies on the bioavailability of polyphenols, the most advantageous forms of administration, the best formulations, and the synergy of polyphenols are necessary in order to clarify the true potential of plant polyphenols in the treatment of metabolic syndrome.

These studies are of significant clinical importance because they may lead to the development of the so-called functional food, which food, unlike typical pharmaceuticals, can contribute to the improvement in the health of the population by influencing common metabolic disorders [110].

## Figures and Tables

**Table 1 biology-11-00559-t001:** The characterization of metabolic changes in genetic animal models of metabolic syndrome.

Strain	Mutation/Genetic Background	Metabolic Changes	Model	References
Zucker Fatty rats (ZF)	missense mutation on the leptin receptor gene (fa/fa)	1. obesity 2. hypertension 3. hyperinsulinemia 4. insulin resistance 5. hypercholesterolemia 6. hypertriglyceridemia	obesity, type II diabetes, MetS	[51,52]
Zucker Diabetic Fatty rats (ZDF)	non-functional leptin receptor (selective in-bread rat strain)	1. obesity 2. hypertension 3. hyperinsulinemia 4. insulin resistance 5. hyperglycemia 6. hypercholesterolemia 7. hypertriglyceridemia	type I and II diabetes, MetS	[49,51,75,76,77]
Spontaneously Hypertensive rats (SHR)	-	1. hypertension 2. hyperinsulinemia 3. insulin resistance	hypertension, heart failure, renal dysfunction	[49,51]

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
