# Peer review of "Biological Potential of Polyphenols in the Context of Metabolic Syndrome: An Analysis of Studies on Animal Models"

_biology, 2022, doi:10.3390/biology11040559_

Round 1
Reviewer 1 Report
The authors have fulfilled all my previous requests. Now the manuscript is more understandable. Minor point: reference 17 please correct the author's name (the correct name is Giacomelli L. instead of Giacomell L.)
Author Response
I am delighted that my correction following your suggestions is acceptable and makes my manuscript more understandable for readers. Also, I am aware that the theme of natural antioxidants in the treatment and prevention of metabolic syndrome is complicated in comprehensive presentation. The amount of study still appearing and changing previous concepts make it almost impossible to present the core of the problem. I corrected the author's name in reference no 17, and I also want to apologize for making such a mistake.
Reviewer 2 Report
The paper by Niewiadomska entitled "Biological potential of polyphenols in the context of metabolic 2 syndrome - analysis of studies in animal models" deals with a very interesting topic.
Obesity and subsequent metabolic syndrome affect millions of people around the world, not only in developed countries, but also in developing countries, where even the quality of ingested food tends to be low (consider the high adoption of processed, high-calorie foods). For this reason, the search for new natural adjuvants that can counteract insulin resistance syndrome is highly recommended.
Niewiadomska's proposal is interesting in terms of its general objective, but it is of very low quality.
According to my observation:
In the bstract:
MetS is not “a health condition”; it is a disease condition;
Polyphenols should be considered as an adjunct to antidiabetic medications;
Better define the objective of the work;
In the main text:
Revise the mechanistic details of metabolic syndrome pathogenesis in updated form;
Explain why it is important to focus on animal models of the disease.
Address and describe more systematically the major risk factors for metabolic syndrome;
Explain better the role of the renin-angiotensin-aldosterone system in this context. Why do you introduce polyphenols in this paragraph?
Write a paragraph chemically describing polyphenols and their related food sources;
Better describe the bibliographic search criteria, the number of papers found, and the selection criteria;
The organization of the manuscript is poor: please describe the goal of the manuscript and how it is organized (by food source?).
Better define the conclusion with a clear clinical message: are there polyphenols that are more active and promising?
What does this work add to the previously published literature?
In general, the paper needs an extensive revision of the English
Author Response
I apologize for saying that metabolic syndrome is a "health condition". It was my mistake because you are right, and I was supposed to have written "disease condition" if so, and I updated it in the corrected abstract. It is also true that polyphenols may be considered a supplement of antidiabetic drugs and may amplify their effect. However, patients with diagnosed metabolic syndrome do not necessarily develop diabetes melitus requiring the administration of drugs, especially metformin. Therefore I focused on the usefulness of polyphenols mainly in the prevention and progression retardation of MetS. In the updated abstract, I added that polyphenol-rich foods should also be considered a supplement of the antidiabetic drugs, following your suggestion because I think it is worth mentioning. I tried to explain the objectives in the abstract better, and I hope I clarified the background of my work.
When it comes to the main part of the text, I revised the MetS pathogenesis. However, I found it rather tricky due to complex correlations between risk factors of MetS and the fact that I want to just put a light on it instead of going deep and presenting the whole spectrum of its pathophysiology.
I corrected the part about the most significant risk factors in MetS, but I am not sure if now is more straightforward for readers. If it still needs a correction, I will try to do my best to describe it more systematically and understandable.
The paper is organized by animal models mimicking in the best way components of MetS. For each animal model, the most promising substances were chosen. Therefore, each main paragraph follows the description of animal models.
I believe that appropriate animal models are crucial in better understanding the pathogenesis of MetS. Moreover, they allow for investigating the impact of new compounds on a morphological, histological, biochemical, and functional level. In contrast, such a broad scope of monitoring is practically impossible in human studies. I explained it in an updated conclusion.
According to your guide, I also discussed in more detail the role of renin-angiotensin-aldosterone axis activation in the context of MetS pathologies. Polyphenols were presented in this paragraph erroneously. Recently I created a separate section introducing polyphenols as chemical substances. Also, I improved the methodological section and completed it with the information suggested by you in the review report form.
I wrote a whole new conclusion part. Unfortunately, the straightforward clinical message is not so apparent to indicate. I summed up the influence of described polyphenols in mitigating pathologies associated with MetS. Nevertheless, further studies seem to be requisite.
In my opinion, this paper tries to present the promising polyphenols in the context of animal models showing the spectrum of changes occurring in MetS. It is not only focused on polyphenols in general but also disputes its validity of usefulness in preventing and relieving metabolic alternations or even its reversal. However, it is still vital to carry out randomized or cohort trials on humans.
I am also aware that I am not an English native speaker, and my writing style requires extensive revision, so I decided to use the editing system offered by the journal. I hope that after the correction, my review will be more acceptable.
Reviewer 3 Report
This review is well designed and written even if this reviewer thinks it needs few minor revisions upon publication.
The authors describe metabolic syndrome which is a combination of several diseases: obesity, type 2 diabetes, insulin resistance and others. They should add a paragraph on cardiovascular disease such as atherosclerosis in order to better introduce all the nutraceuticals they are to going to explain in the paper.
Author Response
Following your advice, I added a subsection entitled: cardiovascular consequences. This paragraph comprises a brief introduction to the influence of metabolic syndrome on the cardiovascular system, including heart failure, microvascular injury, impaired endothelial function, atherosclerotic disease and vascular calcification. As each component of Metabolic Syndrome is an independent risk factor for cardiovascular diseases, it is essential to highlight the cardiovascular pathological changes to understand better the polyphenol's mechanistic target that may mitigate the adverse outcomes of the Metabolic Syndrome.
Round 2
Reviewer 2 Report
I would like to congratulate the authors for responding to all my comments. The changes made have improved the paper. However, in my opinion, the paper is still far from the publication standards for Biology.
Despite the changes, the authors did not define the objectives and did not explain in the introduction why it is important to focus on animal models of the disease. Bibliographic search criteria have been added but could be improved. It is still not clear how authors select and discard papers. The conclusion lacks a clear clinical statement and the authors should explain what this work adds to the already published literature.
Considering the opinion of the other two reviewers, I defer to the editor the acceptance of the manuscript as it is.
Author Response
Here I would like to present the answers to suggestions made by the reviewer. I hope that the corrected version meets the requirements for publishing in Biology.
“I would like to congratulate the authors for responding to all my comments. The changes made have improved the paper. However, in my opinion, the paper is still far from the publication standards for Biology.”
- The authors are satisfied with the current assessment of the manuscript by the reviewer. Extensive improvements were introduced into the new version of the manuscript.
“Despite the changes, the authors did not define the objectives and did not explain in the introduction why it is important to focus on animal models of the disease.”
- According changes were implemented in the manuscript.
“Bibliographic search criteria have been added but could be improved. It is still not clear how authors select and discard papers.”
- The required changes were performed.
“The conclusion lacks a clear clinical statement and the authors should explain what this work adds to the already published literature.”
- The part of the manuscript was changed accordingly.
This manuscript is a resubmission of an earlier submission. The following is a list of the peer review reports and author responses from that submission.
Round 1
Reviewer 1 Report
Polyphenols are organic compounds containing mutiple phenyl hydroxyl groups with anti-oxidative and free radical clearance activities. This paper is a brief review of the biological activities of several polyphenols and polyphenol rich plants. Some important polyphenols, such as anthocyanins were missed.
Reviewer 2 Report
1. The article did not answer the topic: polyphenols to metabolism syndrome.
The authors seemed not understand well the definition of metabolic syndrome and they did not point out the mechanism of polyphenols to metabolic syndrome.
2.In addition, the evidance base studies did not tell us the inclusion and exclusion criteria clear and make the results clear.
Reviewer 3 Report
The manuscript by Joanna Niewiadomska et al., addresses an interesting and current problem about the role of biological potential of polyphenols in the context of metabolic syndrome. However, I have certain suggestions related to the paper. First of all are not present numbered lines in the manuscript and this makes the review little bit complex…
- Please use progressive number for references, some references are repeated, I suggest finding new references instead to repeat the same so thus to enrich the manuscript.
- I suggest to make a unique paragraph for introduction, it is not necessary to split it.
- The introduction is little bit confused and sometimes seems a kind of patchwork and the aim of the study is not very clear. Please rewrite some parts of the introduction in order to make it easy to understand.
- In the methods section please add at least one reference for PRISMA checklist.
- Please add “)” at the end of Methods section: “…independent researchers (JN and AGM”. However the methods section is too confused and must be rewritten.
- In the result section (3.1): “Studies using this animal model have shown that many polyphenolic substances, in the form of extracts or individual compounds, have potential, beneficial metabolic effects”. Please add one or more references!
- In the Introduction or result section I suggest to add this interesting reference by Metere et al. (Absorption, metabolism and protective role of fruits and vegetables polyphenols against gastric cancer, Eur Rev Med Pharmacol Sci. 2017 Dec;21(24):5850-5858. doi: 10.26355/eurrev_201712_14034). It is an interesting review that you can use to enrich your manuscript.
In conclusion, I think the manuscript is a sort of review, but it was submitted as article, and the research was made as meta-analyses. I think the manuscript could be considered a review. However, a lot of data for the “results” and only few lines for “conclusions”. The Conclusion section is poor and must be enriched a lot to consider the manuscript acceptable.
Please in the new submission use numbered lines!